# Antimicrobial Spectrum of Titroleane™: A New Potent Anti-Infective Agent

**DOI:** 10.3390/antibiotics9070391

**Published:** 2020-07-08

**Authors:** Bianca Johansen, Raphaël E. Duval, Jean-Christophe Sergere

**Affiliations:** 1SETUBIO SA, Bioparc Vichy, F-03270 Hauterive, France; Johansen.biancaA@gmail.com; 2Université de Lorraine, CNRS, L2CM, F-54000 Nancy, France; raphael.duval@univ-lorraine.fr

**Keywords:** Titroleane™, Tea Tree oil, antimicrobial activity, bacteria, fungi

## Abstract

Tea Tree oil (TTO) is well known for its numerous good properties but might be also irritating or toxic when used topically or ingested, thus limiting the number of possible applications in Humans. The aim of the study was to characterize the antimicrobial spectrum as well as the toxicity of Titroleane™, a new anti-infective agent obtained from TTO but cleared of its toxic monoterpenes part. The susceptibility to Titroleane™ of various pathogens (bacteria and fungi) encountered in animal and human health was studied in comparison with that of TTO. Antimicrobial screening was carried out using the broth microdilution method. Activities against aerobic, anaerobic, fastidious and non-fastidious microorganisms were performed. For all microorganisms tested, the MIC values for Titroleane™ ranged from 0.08% to 2.5%, except for *Campylobacter jejuni*, and *Aspergillus niger*. In particular, Titroleane™ showed good efficacy against skin and soft tissue infection pathogens, such as methicillin resistant *Staphylococcus aureus* (MRSA), intra-abdominal infections and oral pathogens, as well as fish farming pathogens. Toxicity testing showed little and similar cytotoxicities between TTO and Titroleane™ of 37% and 23%, respectively at a concentration of 0.025% (*v*/*v*). Finally, we demonstrated that the antimicrobial activity of Titroleane™ is similar to that of TTO.

## 1. Introduction

Over the past decade, many investigations have supported the activity of Tea Tree oil (TTO) against microbes and parasites [1,2,3]. The antimicrobial activity of TTO has been attributed to several molecules, such as terpinen-4-ol, α-terpineol, linalool, α-pinene, β-pinene and 1,8-cineole, as well as α-terpinene, γ-terpinene and p-cymene as antibacterial molecules and terpinolene as antifungal [3,4,5,6,7,8,9,10,11,12,13,14].

However, some of these molecules (or their oxidation product) are known to be toxic and irritant and are considered as allergens. This includes terpinolene, γ-terpinene, α-pinene, 1,8-cineol and p-cymene [15,16]. Because of these monoterpenes, batches of TTO can have very different toxicological profiles [17]. In order to regulate the quality of TTO, the international Standard ISO 4730, describes the requirements in the amounts of the different compounds. 

A proprietary owned process of the fractionation of SETUBIO led to a derivative TTO named Titroleane™, by releasing the monoterpenes contained in TTO [18]. It is enriched in active molecules, such as monoterpenes alcohol—previously identified by Southwell et al. [19] and Carson and Riley [20]—at a guaranteed concentration >60%, with a minimal monoterpenes concentration of <5% instead of 40% in standard TTO. In addition to the production of Titroleane™, a second fraction, called Fraction 2, is generated, mostly composed of monoterpenes. The aim of the study is to evaluate the antimicrobial spectrum of Titroleane™ and to validate that, even with the removal of toxic and irritating molecules (also known for their antimicrobial properties), the activity of Titroleane™ is similar to that of TTO. Industrial applications and other potential properties of Titroleane™ will also be discussed.

## 2. Results

### 2.1. Comparative Compounds of Standard TTO and Titroleane™ 

Table 1 shows the chemical composition of TTO, Titroleane™ and Fraction 2 obtained by Gas Chromatography with Flame Ionization Detector (GC-FID). TTO and Fraction 2 contain high amounts of monoterpenes with a maximal concentration of 72% and 83%, respectively, while Titroleane™ contains less than 3.2%. Conversely, monoterpenes alcohols are present in higher concentrations in Titroleane™ compared to TTO and Fraction 2. Indeed, Titroleane™ contains 71 to 75% of terpinen-4-ol, instead of 48% in TTO and 9.6% in Fraction 2. Titroleane™ also contains 5% to 9% of α-terpineol, while standard TTO never contains more than 5%. Concerning sesquiterpenes and sesquiterpenes alcohols in Titroleane™ and TTO, both rates are low—viridiflorene is present at 3.3% in Titroleane™ instead of 3% in TTO. Similarly, δ-cadinene represents 2.8% in Titroleane™ and 3% in TTO. Fraction 2 contains less than 1% α-terpineol and less than 0.01% sesquiterpenes and sesquiterpenes alcohols.

Results in percentages represent the highest and the lowest concentrations of main compounds. The chromatographic profile of standard Tea Tree oil is set by the norm ISO/FDIS 4730:2017. Concentrations in Titroleane™ result from three independent batches and from one batch for Fraction 2. Analyses were performed by GC-FID.

We also wanted to determine the impact of the composition of each extract on plastic containers. Appendix A shows the impact of extract composition on plastic containers after two-month storage. TTO and Titroleane™ (whose main compound is terpinen-4-ol) have regular containers, while Fraction 2 shows a modified plastic container. This behavior must be due to its high concentration in monoterpenes.

### 2.2. Homogeneity between Batches

As GC-FID analyses on three productions of Titroleane™ have demonstrated similar compositions (data not shown), we also wanted to test the homogeneity of our process by evaluating the antibacterial activity of different batches. Consequently, four different and independent batches (i.e., B1, B2, B3 and B4) were tested for their antibacterial activity against *Escherichia coli*, *Staphylococcus aureus* and *Yersinia enterocolitica*. They demonstrated the same antibacterial activity—the susceptibility of *E. coli*, *S. aureus* and *Y. enterocolitica* to the different productions of Titroleane™ ranged from 2.5% to 1.25%, as shown in Table 2.

### 2.3. MIC of Titroleane™ and Fraction 2 

Titroleane™ and Fraction 2 were tested in MIC assays for their respective antibacterial activities. Regarding the production batch B2 in Table 3, the susceptibility of *E. coli*, *S. aureus*, and *Y. enterocolitica*, were 1.25% and 2.5% for Titroleane™ and Fraction 2, respectively. Similar results were measured for production batch B3, with MIC at 2.5% for both fractions on *S. aureus* and *Y. enterocolitica*. *E. coli* was susceptible to 2.5% Titroleane™ and no MIC was recorded for Fraction 2.

### 2.4. Antibacterial Spectrum of Titroleane™

MICs were performed on forty-nine bacteria and five fungi to define the antimicrobial spectrum of Titroleane™. There was no clear trend of inhibition depending on the characteristics of the bacterial strains, as shown in Table 4. Regarding Gram-positive bacteria, Titroleane™ showed good activity toward Bacilliales and Lactobacilliales, with MIC ranging from 0.62% to 2.5% and from 1.25% to 2.5%, respectively. Similar results were obtained for *C. xerosis* (MIC = 1.25%), and *Clostridium* sp. (MIC = 0.62% to 2.5%). Titroleane™ showed good efficiency on wild type *S. aureus* and *E. coli* (MIC = 0.62%), as well as against MRSA (Methicillin-Resistant *S. aureus*) (MIC = 1.25%). Considering the inhibition of spore germination, Titroleane™ had a good anti-germination effect on *B. atrophaeus* spores at 0.62%. Within Gram-negative bacteria, Enterobacteriales, which are of higher concern in threats, were the most sensitive to Titroleane™ (MIC = 0.31% to 1.25%). Bacteroidales were also very susceptible to Titroleane™, mostly *B. fragilis* with a MIC at 0.08%. Titroleane™ also showed antifungal activity against *Candida* sp. (MIC = 1.25%), and the mould *M. furfur* (MIC = 1.25%). No activity was observed against *A. niger* at tested concentrations.

### 2.5. Activity of Titroleane™ versus TTO 

To compare the activity between Titroleane™ and TTO, another series of MICs was performed on two fungi and thirty-one bacteria representing a wide variety of microorganisms. TTO and Titroleane™ showed similar activities toward tested bacteria and yeast, as shown in Table 5. Their MICs only differed for *Y. enterocolitica* (0.31% TTO versus 1.25% Titroleane™), *V. anguillarum* (0.15% TTO versus 0.62% Titroleane™) and *V. nigripulchritudo* (0.15% TTO versus 1.25% Titroleane™). 

### 2.6. Cytotoxicity of TTO versus Titroleane™

The cytotoxicity of Titroleane™ was evaluated on human fibroblast in comparison to TTO, as shown in Figure 1. A low and stable concentration of solvent (DMSO) was used to minimize the impact on cell viability. In the same way, extracts were tested at low concentrations, ranging from 0.025% to 0.006%. The control SDS shows significant cytotoxic activity at a rate of 98%. Little toxicity was recorded with a maximum of 37% with TTO and a significant 23% with Titroleane™ at a concentration of 0.025%. No cytotoxicity was measured at 0.006% for both extracts.

## 3. Discussion

Novel antimicrobial agents are still needed to counteract infections and, in the last 20 years, the search for novel antimicrobial agents from plants has been of great interest [21].

Facing the worldwide dissemination of resistant pathogens, as well as the lack of new therapeutic options, it is now urgent to discover new active antimicrobial agents to fight and treat resistant infections [22]. In this context, during the last 20 years we have observed a rekindling interest of the scientific community in exploring the plant world in order to find these new antimicrobial agents [23]. Nevertheless, essential oils can also be dangerous because of their composition in active but also toxic molecules. Indeed, standard TTO is well known for its numerous good properties but might be also irritating or toxic when used topically or ingested, thus limiting the number of possible human applications [15,16,24,25].

SETUBIO, the original process of detoxifying Tea Tree essential oil, generates two fractions: one, named Titroleane™, in which bioactive beneficial compounds of TTO are concentrated; a second, identified as Fraction 2, which mainly contains monoterpenes. Indeed, gas chromatography analysis of Titroleane™ showed an important reduction of its monoterpenic content compared to TTO, as shown in Table 1. Titroleane™ contains 100-times less α -pinene, more than five-times less terpinolene, 10-times less 1,8-cineole, 37-times less α-terpinene, 10-times less γ-terpinene and 26-times less limonene than TTO. On the contrary, Fraction 2 contains no sesquiterpenes nor sesquiterpenes alcohol, and almost two-fold more γ-terpinene than TTO. Even if TTO and Titroleane™ have quite different compositions, their cytotoxicity on fibroblasts is similar. Titroleane™ is, nevertheless, of interest since it is clear of several known allergens [15,16].

The chromatographic profiles were studied on different batches, and both fractions had constant activity toward tested strains, which attest to an accurate production process for Titroleane™. The similar antimicrobial activity observed for fractions is supported by research pursuing isolated single compounds of TTO. First, 1,8-cineol, has been described widely for its antibacterial activity on vegetative bacteria, biofilm [26,27] or fungi [28,29]. Second, α-pinene has also been studied for its antibacterial activity on Gram-negative bacteria and *C. albicans* [10,30,31,32]. Third, γ-terpinene has initially been reported to be inactive [20,33], but recent reports described antimicrobial activity and explored its mechanism [34,35]. Fourth, cymene was found to potentiate the antimicrobial activity of other molecules and might indirectly play a role in the antibacterial activity of Fraction 2 [12].

Terpinen-4-ol has been identified in several studies as the principal active component of TTO with α-terpineol, effective, among others, against bacterial skin pathogens [19,20,27,30,36,37]. Terpinen-4-ol was defined as the best antifungal component of TTO, followed by α-terpineol [6]. These two compounds represent more than 70% of the composition of Titroleane™, which could explain the very good antimicrobial activity of the product.

To define the antimicrobial spectrum of Titroleane™, various microorganisms were tested. As shown in Table 4, Titroleane™ has a sporicidal activity against *B. atrophaeus* and is active on *C. sporogenes*, *Y. enterocolitica*, *V. cholerae* and *S. enterica enterica typhimurium*. They all represent non-toxigenic surrogate strains involved in bioterrorism and epidemic infection, such as anthrax, botulisms, and plague [38].

Gram-negative bacteria, which are of high concern in threats, showed good susceptibility to Titroleane™. Indeed, its antibacterial activity is particularly pronounced against Enterobacteriaceae, such as *E. coli* and *S. enterica enterica typhimurium*, which are responsible of half of the community acquired intra-abdominal infections in the European Union, according to Sartelli M and collaborators [39]. Good efficacy against *Bacteroides* spp., which is the most represented anaerobic bacteria in Europe for intra-abdominal infection [39], has also been demonstrated. Unfortunately, bacteria involved in disease prevention, such as *Lactobacillus* spp. and *Bifidobacterium* spp. [40,41], are also susceptible to Titroleane™. Nevertheless, depending on the concentration used, Titroleane™ could kill or inhibit pathogens or potential pathogens with a limited impact (or side effect) on the commensal ones. This has been previously demonstrated for TTO [42].

Many pathogens involved in skin infection have demonstrated interesting susceptibilities to Titroleane™, such as *S. aureus* and MRSA, *S. epidermidis*, *E. faecalis*, *Klebsiella*. spp., *C. albicans*, and *P. aeruginosa*. These pathogens are responsible for more than 50% of the hospital acquired surgical site infections (SSI) in the US, according to the BiomedTracker Part 1 report of September 2014 [43], and 100% of the French community’s acquired skin and soft tissue infections (SSTI), according to the national survey on nosocomial infections in 2012 [44]. 

Regarding its antibacterial spectrum, we have demonstrated that Titroleane™ is also active against periodontal bacteria [45,46], involved in caries and plaque formation, as well as against acne vulgaris [47] and vaginosis, caused by *Candida* spp. [48]. Titroleane™ might also be of interest in non-cholerae vibriosis infections, common in fish farming and with a rising incidence in humans [49].

One could expect that the antimicrobial activity spectrum of Titroleane™ is weaker than those of standard TTO, due to the absence of numerous compounds in Titroleane™ that could potentiate its antibacterial activity. Indeed, it has been found that terpinen-4-ol and α-terpineol have an antagonist effect in killing demodex mites, whereas terpinen-4-ol and terpinolene have a synergistic effect. These data would suggest that Titroleane™ could be less active than TTO, by the absence of a synergistic effect brought about by the presence of terpinolene [14]. Surprisingly, as demonstrated by the results in Table 5, Titroleane™ has the same efficacy as TTO.

TTO’s MIC values presented in this study are supported by many previous articles. Its sporicidal activity against *B. subtilis* spores at a concentration above 1% has been previously demonstrated [50], as well as its activity against oral pathogens at 0.2% [51]. Several studies also showed its activity against enteric and skin pathogens [52,53] and in vivo testing demonstrated the activity of TTO against bacteria involved in acne [54]. However, previous experiments on *P. aeruginosa* showed no antibacterial activity of TTO, whereas in the present study, we clearly demonstrated that *P. aeruginosa* is sensitive. This discrepancy can be explained by the different strains used and/or the methods used in the preparation of the oil [53].

Because Titroleane™ has similar MICs to TTO, one can reasonably believe that it could be studied for the same applications. 

Concerning periodontal disease, a TTO gel was found to have a positive effect on chronic periodontitis after local delivery [55]. In addition, an evaluation of the effect of a 0.2% TTO mouthwash on the oral flora of forty volunteers suggests that TTO could reduce the total number of oral bacteria and decrease the dental biofilm [56,57]. Regarding this good efficiency to inhibit the growth of periodontal strains, Titroleane™ could also been used for mouth hygiene, with less known molecules than TTO that are toxic to human health.

Concerning skin infections, wound dressing has been formulated with TTO and studies suggest that it promotes healing as well as the decolonization of *S. aureus* of human wounds in vivo [58,59], probably because of terpinen-4-ol and α-terpineol, which are able to penetrate the entire thickness of the epidermis [60]. The application of dressing made from Titroleane™ could be used to control SSTI infections in hospitals and in the community. 

Numerous formulations and studies were made on the effectiveness of TTO in acne treatment in vitro and in vivo [61], thus, the results of Titroleane™ look promising toward a future treatment for acne.

The activity of compounds shared by TTO and Titroleane™ suggest other potential properties for Titroleane™, such as antiparasitic activity [62,63,64,65] and acaricidal activity [66,67], as well as virucidal activity [68,69]. Both terpinen-4-ol and α-terpineol were found to be immunomodulators, by reducing anti-inflammatory response, and were found to have antioxidant properties [70,71,72,73].

Studies on TTO compounds suggest that many more properties than its broad spectrum of antimicrobial activity could be allocated to Titroleane™. Further investigations have to be done to fully discover the whole potential of Titroleane™ for human health benefits.

Finally, the toxicity of TTO and Titroleane™ was investigated. The difference between both oils is not as high as we expected but can be explained in two ways. The first reason is in the model used. Half of molecules in TTO are hydrophobic, whereas most of the Titroleane™ molecules are hydrophilic. This means that, in this model (as in most cellular culture models), more molecules are solubilized and in contact with the cells with Titroleane™ than with TTO. In other words, molecules of Titroleane™ must be more accessible to cells than those of TTO. The second reason is that Titroleane™ is a concentrate of TTO’s monoterpene alcohol fraction. In this test, however, we used the same concentrations to compare them. The results show that the cytotoxicity of Titroleane™ and TTO are similar, even if their compositions are highly different.

## 4. Materials and Methods 

### 4.1. Chemicals

#### 4.1.1. Active Compounds

Standard Tea Tree Essential Oil (TTO), according to the norm ISO/FDIS 4730:2017 was purchased from Helpac (Auzon, France). The compound profile of Titroleane™ was obtained from Lexva Company (Saint-Beauzire, France), by gas chromatography with flame ionization detector (GC-FID).

#### 4.1.2. TTO Fractionation

TTO’s fractionation was performed by distillation. TTO was loaded on a fractionating Vigreux-type column (with finger indentations) of 3 m high and 10 cm diameter, at 72 °C at the head of the column. The distillation of TTO was used to separate the mixture into two fractions, based on their volatilities. The fraction containing sesquiterpenes and alcohols was named Titroleane™, while the fraction mostly composed of monoterpenes was named Fraction 2.

Batches were stored in 10 L or 20 L high density polyethylene containers. Samples were collected in 60 mL polystyrene flasks and stored at 22 °C and protected from light. In order to evaluate the impact of the composition on plastic containers, 60 mL polystyrene flasks were observed after two months. Batch B2 and associated TTO and fraction 2 were used for the visual observation.

#### 4.1.3. Industrial Batches

Four industrial batches of Titroleane™, named B1, B2, B3 and B4, were produced at different times. If not specified, Titroleane™ refers to batch B2. The antimicrobial efficacy of these batches was tested on an initial set of three bacterial species: *Escherichia coli* as a Gram-negative model, *Staphylococcus aureus* as a Gram-positive model and *Yersinia enterocolitica* as a foodborne model. Assays were performed once immediately after production.

#### 4.1.4. Samples Preparation

To enhance the solubility of TTO and Titroleane™, Dimethyl sulfoxide (DMSO) (Sigma Aldrich, Saint-Quentin-Fallavier, France) was used as a solvent. Oils were diluted at 10% final volume (*v*/*v*) in 10% DMSO. All experiments were performed simultaneously with a control solution of 10% DMSO.

### 4.2. Antimicrobial Activities

#### 4.2.1. Bacteria and Fungi

*Bacillus cereus* (ATCC 11778), *Listeria monocytogenes* (ATCC 19115), *Enterococcus hirae* (ATCC 8043), *Klebsiella oxytoca* (ATCC 49131), *Yersinia enterocolitica* (ATCC 23715), *Corynebacterium xerosis* (ATCC 7711), *Vibrio anguillarum* (ATCC 19264), *Vibrio parahaemolyticus* (ATCC 17802), *Campylobacter jejuni* (ATCC 29428), *Propionibacterium acnes* (ATCC 6919) and *Candida kefyr* (ATCC 2512) were purchased from the American Type Culture Collection (ATCC).

*Bacillus subtilis* (CIP 52.62), *Bacillus atrophaeus* (CIP 107159), *Staphylococcus aureus* (CIP 4.83), Methicillin-resistant *Staphylococcus aureus* MRSA (CIP 107422), *Staphylococcus epidermidis* (CIP 68.21), *Enterococcus faecalis* (CIP 103214), *Enterococcus faecium* (CIP 102379), *Salmonella enterica enterica typhimurium* (CIP 103799), *Escherichia coli* (CIP 53.126), *Shigella boydii* (CIP 52.48), *Vibrio cholerae* (CIP 106973), *Vibrio vulnificus* (CIP 103196), *Acinetobacter baumannii* (CIP 70.34), *Pseudomonas aeruginosa* (CIP 82.118), *Vibrio alginolyticus* (CIP 103336), *Vibrio harveyi* (CIP 104172), *Vibrio nigripulchritudo* (CIP 103195), *Actinomyces naeslundii* (CIP 100654), *Helicobacter pylori* (CIP 103995), *Fusobacterium nucleatum* (CIP 101130), *Bacteroides fragilis* (CIP 105891), *Bacteroides ovatus* (CIP 103756), *Bacteroides thethaiotaomicron* (CIP 104207), *Bifidobacterium bifidum* (CIP 56.7), *Bifidobacterium breve* (CIP 64.69), *Bifidobacterium lactis* (CIP 105265), *Bifidobacterium longum longum* (CIP 64.62T) and *Bifidobacterium longum infantis* (CIP 64.71) were purchased from “Collection de l’Institut Pasteur”(CIP, Pasteur Institute Collection).

*Candida albicans* (UMIP 48.72), *Candida tropicalis* (UMIP 2148.93), *Aspergillus niger* (UMIP 1431.83) and *M. furfur* (UMIP 1634.86) were purchased from UMIP (Pasteur Institute Collection of fungi).

Master stocks of bacterial strains and fungi were stored in a −80 °C freezer. 

#### 4.2.2. Growth Conditions and Inoculum Preparation

The bacteria and fungi were grown on the plate and then a single colony was transferred to a second plate. For aerobic bacteria, incubation time was 24 h, for fungi and anaerobic bacteria the time was 48 h and for microaerobic strains the time was 72 h. The oxygen-free atmosphere was obtained by using GenBOX™ Anaer or Microaer (bioMérieux, Craponne, France). The inoculum was prepared from isolates of the second agar plate in TPS solution (Tryptone 0.1% (211705, BD, Le Pont de Claix, France) and Sodium chloride 0.85% (S9888, Sigma Aldrich, Saint-Quentin-Fallavier, France)), and was then diluted in adequate broth at 5.10^5^–1.10^6^ CFU mL^−1^. 

All strains were cultivated at 37 °C except for moulds and yeasts, which were grown at 30 °C, as well as *B. cereus*, *B. atrophaeus*, *S. boydii*, *V. cholerae*, *V. harveyi*, *V. vulnificus*, *A. baumannii* and other *Vibrio* spp., which were grown at 22 °C.

Trypticase soy agar and broth (236920 and 211825, BD, Le Pont de Claix, France) were used to grow *Bacillus cereus* (ATCC 11778), *Bacillus subtilis* (CIP 52.62), *Bacillus atrophaeus* (CIP 107159), *Staphylococcus aureus* (CIP 4.83), Methicillin-resistant *Staphylococcus aureus* (MRSA) (CIP 107422), *Staphylococcus epidermidis* (CIP 68.21), *Listeria monocytogenes* (ATCC 19115), *Enterococcus faecalis* (CIP 103214), *Enterococcus faecium* (CIP 102379), *Enterococcus hirae* (ATCC 8043), *Klebsiella oxytoca* (ATCC 49131), *Salmonella enterica enterica typhimurium* (CIP 103799), *Escherichia coli* (CIP 53.126), *Shigella boydii* (CIP 52.48), *Yersinia enterocolitica* (ATCC 23715), *Vibrio cholerae* (CIP 106973), *Vibrio vulnificus* (CIP 103196), *Acinetobacter baumannii* (CIP 70.34), *Pseudomonas aeruginosa* (CIP 82.118) and *Corynebacterium xerosis* (ATCC 7711). 

Trypticase soy agar and broth +2% NaCl was used to grow *Vibrio alginolyticus* (CIP 103336), *Vibrio anguillarum* (ATCC 19264), *Vibrio harveyi* (CIP 104172), *Vibrio nigripulchritudo* (CIP 103195) and *Vibrio parahaemolyticus* (ATCC 17802). 

Columbia agar and broth (AEB151002N, AES Laboratoire, Combourg, France and 294420, BD, Le Pont de Claix, France) were used to grow *Actinomyces naeslundii* (CIP 100654). 

Columbia +5% sheep blood agar (bioMérieux, Craponne, France) and Columbia (BD, Le Pont de Claix, France) +10% horse serum (16050, Gibco, Paisley, UK) were used to grow *Campylobacter jejuni* (ATCC 29428) and *Helicobacter pylori* (CIP 103995).

Medium for anaerobic bacteria was made according to the Pasteur Institute medium 20 protocol: Tryptone (211705, BD, Le Pont de Claix, France); Yeast extract (1196–5365, Fisher Scientific, Villebon-sur-Yvette, France); L-caseine hydrochloride (C7880, Sigma-Aldrich, Saint-Quentin-Fallavier, France); hemine (25 mg mL^−1^); Glucose; pastagar B 15 g/L (AEB175356, AES Laboratoire, Combourg, France). 

Solution Hemine 25 mg mL^−1^ (Hemine chloride (H9039, Sigma-Aldrich, Saint-Quentin-Fallavier, France) and Triethanolamine (421631000, ACROS Organics, Antwerp, Belgium)) were used to grow *Fusobacterium nucleatum* (CIP 101130), *Bacteroides fragilis* (CIP 105891), *Bacteroides ovatus* (CIP 103756), *Bacteroides thethaiotaomicron* (CIP 104207), *Bifidobacterium bifidum* (CIP 56.7), *Bifidobacterium breve* (CIP 64.69), *Bifidobacterium lactis* (CIP 105265), *Bifidobacterium longum longum* (CIP 64.62T), *Bifidobacterium longum infantis* (CIP 64.71) and *Propionibacterium acnes* (ATCC 6919). 

Sabouraud dextrose agar and broth (210950 and 238230, BD, Le Pont de Claix, France) were used to grow *Candida albicans* (UMIP 48.72), *Candida kefyr* (ATCC 2512), *Candida tropicalis* (UMIP 2148.93) and *Aspergillus niger* (UMIP 1431.83). 

Sabouraud dextrose agar and broth + sterile olive oil were used to grow *M. furfur* (UMIP 1634.86).

#### 4.2.3. *Bacillus atrophaeus* Spore’s Suspension Preparation

Plates with isolated colonies were incubated for two weeks at 37 °C, then used to prepare a highly concentrated suspension in TPS solution. The suspension was pasteurized at 80 °C during 20 min to kill all remaining vegetative cells. The number of spores was then counted using a Malassez cell-counting chamber and diluted to 10^6^ spores mL^−1^.

#### 4.2.4. Minimum Inhibitory Concentration (MIC) Determination

MICs values were measured using the microdilution broth method, according to the Clinical and Laboratory Standards Institute guidelines [74], with modification on the broths used to fit with organism requirements for growth. Indeed, some authors have shown that there are no significant differences, depending on the culture medium used (i.e. Sabouraud dextrose broth (SDB), RPMI and Christensen’s urea broth (CUB)) for the determination of MICs using the broth microdilution (BMD) method, recommended by the CLSI [75]. Hence, standard protocol was adapted to fungi as follows: fungi were diluted at 10^5^ CFU mL^−1^ in Sabouraud broth (*Candida albicans*, *Candida kefyr*, *Candida tropicalis*, *Aspergillus niger*) or Sabouraud broth + 20% olive oil (*Malassezia furfur*); plates were incubated aerobically at 30 °C and analyzed after 48 h; then, 96-well microplates were prepared with three controls—growth control without product, negative control without bacteria, and solvent control (DMSO from 2.5% to 0.0049%); wells were prepared by serial dilutions from 2.5% to 0.0049% with 50 µL of product and 50 µL of suspension; microplates were then incubated in growth conditions defined above, then MICs were read visually. The same batch of Titroleane™ was used for the whole spectrum study. A two-fold variation between two MICs, corresponding to one dilution, was considered as non-significant.

### 4.3. In Vitro Cytotoxicity

Human foreskin fibroblast cell lines were purchased from ATCC collection SCRC-1041. Cells were grown in MEM (10370-047, Gibco, Paisley, UK) with the addition of 10% heat-inactivated fetal bovine serum (10270, Gibco, Paisley, UK), 2 mM of L-glutamine (P04-80100, PAN™ biotech, Aidenbach, Germany), 100 µg mL^−1^ ampicillin (A9518, Sigma-Aldrich, Saint-Quentin-Fallavier, France), 0.1 mU mL^−1^ of penicillin–streptomycin (P4333, Sigma-Aldrich, Saint-Quentin-Fallavier, France) and 2.5 µg mL^−1^ of amphotericin B (P06-01050, PAN™ biotech, Aidenbach, Germany). Cells were maintained at 37 °C in a 5% CO_2_ humidified atmosphere. The cytotoxicity was carried out by Neutral Red coloration assay for investigating cell viability [76]. Cells were seeded on 96-well plates, at a concentration of 10^5^–10^6^ cells per mL in completed MEM, and incubated for 24 h. Various concentrations of samples, all prepared in 0.3% DMSO (final concentration in wells *v*/*v*) and sonicated for 10 min before use, were then added to wells with a final volume of 200 µL. Negative (no treatment), solvent (DMSO 0.3%), and positive (0.1% SDS, BP166, Fisher Scientific, Villebon-sur-Yvette, France) controls were included. Each condition was tested three times per assay. Exposure periods of 24 h were chosen for determining and comparing the in vitro cytotoxicity potential of samples. After incubation, the supernatant was removed and cells were washed three times with PBS (10010-015, Gibco, Paisley, UK) before adding Neutral Red (229810250, ACROS Organics Antwerp, Belgium) and solution prepared in completed MEM at 0.005% (*v*/*v*). The plate was then incubated for an additional 3 h. The Neutral Red solution was washed three times in PBS, the coloration was solubilized in acetic acid 1%:ethanol 50% (*v*/*v*) and the absorbance was measured at 540 nm using a microplate reader (Infinite M200 pro, Tecan, Crailsheim, Germany). The cell survival rate was determined by comparing the absorbance values obtained with treated cells and with DMSO. 

### 4.4. Statistical Analysis

The data were analyzed using a one-way ANOVA Dunnet statistical analysis, which allows for the comparison of data to a control—here the solvent control [77]. Statistical analysis was carried out with the free online SAS version (SAS Studio version 9.4), with the use of Oracle VM VirtualBox version 5.1, and VMware Player version 5.0 (SAS University Edition).

## 5. Patents

Patent application was submitted (Application FR2977799–A1).

## Figures and Tables

**Figure 1 antibiotics-09-00391-f001:**
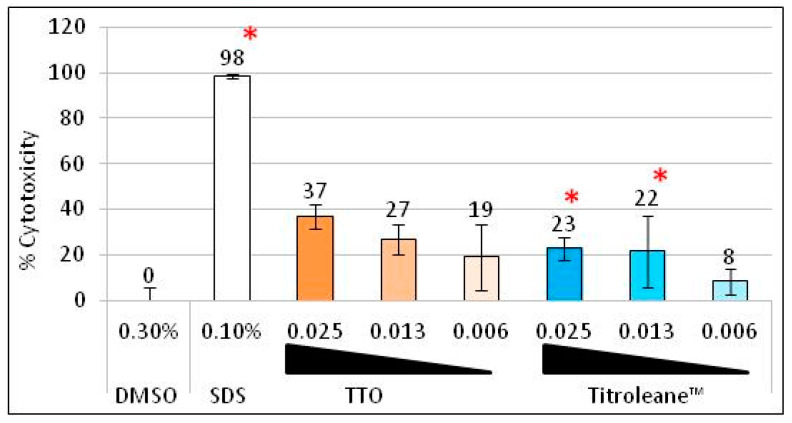
Cytotoxicity of TTO versus Titroleane™. Cytotoxicity percent of TTO and Titroleane™ on human foreskin fibroblast cell lines after 24 h of contact. Extracts are prepared in 0.3% DMSO. Concentrations ranged from 0.025% to 0.006% (*v*/*v*). Experiment was performed three times in triplicates on different days. ***** = *p* < 0.05.

**Table 1 antibiotics-09-00391-t001:** Chromatographic profile of Tea Tree oil, Titroleane™, and Fraction 2.

Terpenes Family	Component	Tea Tree oil (%) Min/Max	Titroleane™ (%) Min/Max	Fraction 2 (%)
Monoterpenes	α-Pinene	1/4	Trace	5.1
	Sabinene	Trace/3.5	Trace	0.2
	α-Terpinene	6/12	0.3/0.4	16.9
	Limonene	0.5/1.5	0.03/0.07	2.9
	p-Cymene	0.5/8	0.2/0.4	8.2
	1,8-Cineol (Eucalyptol)	Trace/10	Trace/0.3	4.2
	γ-Terpinene	14/28	1.7/2.8	40.4
	Terpinolene	1.5/5	0.5/0.8	5.3
Monoterpenes alcohol	Terpinen-4-ol	35/48	71/75	9.6
	α-Terpineol	2/5	5/9	0.2
Sesquiterpenes	Aromadendrene	0.2/3	1.9/3.3	Trace
	Leden (Viridiflorene)	0.1/3	0.9/2.8	Trace
	δ-Cadinene	0.2/3	0.6/2.2	Trace
Sesquiterpenes alcohol	Globulol	Trace/1	0.4/0.9	Trace
	Viridiflarol	Trace/1	0.2/0.4	Trace

**Table 2 antibiotics-09-00391-t002:** Minimal inhibitory concentrations (MICs) of Titroleane™ from four different industrial batches.

Strains	Batch Number	Titroleane^TM^ (%)
*E. coli*	B1	1.25
	B2	1.25
	B3	2.5
	B4	2.5
*S. aureus*	B1	1.25
	B2	1.25
	B3	2.5
	B4	2.5
*Y. enterocolitica*	B1	2.5
	B2	1.25
	B3	2.5
	B4	2.5

B1, B2, B3, B4: four different concentrations industrially produced in January 2012, February 2012, June 2012 and October 2012, respectively. Tests were performed once each.

**Table 3 antibiotics-09-00391-t003:** Minimal inhibitory concentrations (MICs) of two fractions of Tea Tree oil. Titroleane™ and Fraction 2 are complementary fractions of standard Tea Tree oil. Both fractions were compared for their MIC activities on three representative bacterial strains.

	B2 (%)		B3 (%)	
Strain	Titroleane^TM^	Fraction 2	Titroleane^TM^	Fraction 2
*E. coli*	1.25	2.5	2.5	-
*S. aureus*	1.25	2.5	2.5	2.5
*Y. enterocolitica*	1.25	1.25	2.5	2.5

-: no MIC found at tested concentrations (from 2.5% to 0.0049%). Tests were performed once each.

**Table 4 antibiotics-09-00391-t004:** Antimicrobial spectrum of Titroleane™ on bacteria and fungi encountered in human and animal health.

Order	Genus	Species	Titroleane^TM^ (%)
**GRAM-POSITIVE BACTERIA**			
Bacillales	*Bacillus*	*B. cereus*	2.50
		*B. subtilis*	1.25
		*B. atrophaeus*	1.25
		*B. atrophaeus* spores	0.62
	*Staphylococcus*	*S. aureus*	0.62–1.25
		MRSA	1.25
		*S. epidermidis*	0.62–1.25
	*Listeria*	*L. monocytogenes*	1.25
Lactobacillales	*Enterococcus*	*E. faecalis*	2.50
		*E. faecium*	1.25
		*E. hirae*	2.50
	*Lactobacillus*	*L. acidophilus*	1.25
		*L. gasseri*	2.50
		*L. paracasei*	2.50
		*L. rhamnosus*	2.50
	*Streptococcus*	*S. mutans*	1.25
		*S. oralis*	2.50
		*S. sobrinus*	1.25
Selenomonadales	*Veillonella*	*V. dispar*	0.02
Clostridiales	*Clostridium*	*C. difficile*	0.62
		*C. sporogenes*	1.25–2.50
Bifidobacteriales	*Bifidobacterium*	*B. bifidum*	0.62
		*B. breve*	1.25
		*B. longum infantis*	0.62
		*B. longum longum*	0.62
		*B. lactis*	1.25
Actinomycetales	*Actinomyces*	*A. naeslundii*	0.15–0.62
	*Corynebacterium*	*C. xerosis*	1.25
	*Propionibacterium*	*P. acnes*	0.62
**GRAM-NEGATIVE BACTERIA**			
Enterobacteriales	*Klebsiella*	*K. oxytoca*	1.25
	*Salmonella*	*S. enterica enterica typhimurium*	0.62
	*Escherichia*	*E. coli*	0.62–1.25
	*Shigella*	*S. boydii*	0.62
	*Yersinia*	*Y. enterocolitica*	1.25
Vibrionales	*Vibrio*	*V. alginolyticus*	0.62
		*V. anguillarum*	0.62
		*V. cholerae*	1.25
		*V. harveyi*	0.31
		*V. nigripulchritudo*	1.25
		*V. parahaemolyticus*	1.25
		*V. vulnificus*	0.62
Pseudomonadales	*Acinetobacter*	*A. baumannii*	1.25
	*Pseudomonas*	*P. aeruginosa*	1.25–2.50
Campylobacteriales	*Campylobacter*	*C. jejuni*	-
	*Helicobacter*	*H. pylori*	2.50
Fusobacteriales	*Fusobacterium*	*F. nucleatum*	0.019
Bacteroidales	*Bacteroides*	*B. fragilis*	0.08
		*B. ovatus*	0.62
		*B. thetaiotaomicron*	0.31
**FUNGI**			
Saccharomycetales	*Candida*	*C. albicans*	1.25
		*C. kefyr*	1.25
		*C. tropicalis*	0.62–1.25
Eurotiales	*Aspergillus*	*A. niger*	-
Malasseziales	*Malassezia*	*M. furfur*	1.25

-: no MIC found at tested concentrations (from 2.5% to 0.0049%). Screening was performed at least once. MRSA: Methicillin-Resistant *S. aureus*.

**Table 5 antibiotics-09-00391-t005:** Comparison of the minimal inhibitory concentrations (MICs) of Titroleane™ and Tea Tree oil (TTO). They were prepared and tested the same way.

Order	Genus	Species	Titroleane^TM^ (%)	TTO (%)
**GRAM-POSITIVE BACTERIA**				
Bacillales	*Bacillus*	*B. cereus*	2.50	1.25
	*Staphylococcus*	*S. aureus*	0.62–1.25	0.62
		MRSA	1.25	0.62
		*S. epidermidis*	0.62–1.25	0.62
	*Listeria*	*L. monocytogenes*	1.25	1.25
Lactobacillales	*Enterococcus*	*E. faecalis*	2.50	2.50
		*E. faecium*	1.25	2.50
		*E. hirae*	2.50	2.50
	*Lactobacillus*	*L. acidophilus*	1.25	1.25
		*L. casei*	2.50	2.50
		*L. gasseri*	2.50	2.50
		*L. rhamnosus*	2.50	2.50
Clostridiales	*Clostridium*	*C. difficile*	0.62	1.25
		*C. sporogenes*	1.25–2.50	2.50
Bifidobacteriales	*Bifidobacterium*	*B. bifidum*	0.62	1.25
		*B. longum infantis*	0.62	0.62
		*B. longum longum*	0.62–1.25	2.50
		*B. lactis*	1.25	0.62
Actinomycetales	*Corynebacterium*	*C. xerosis*	1.25	1.25
**GRAM-NEGATIVE BACTERIA**				
Enterobacteriales	*Salmonella*	*S. enterica enterica typhimurium*	0.31–0.62	0.31
	*Escherichia*	*E. coli*	0.62–1.25	1.25
	*Yersinia*	*Y. enterocolitica*	1.25	0.31
Vibrionales	*Vibrio*	*V. alginolyticus*	0.62	0.31
		*V. anguillarum*	0.62	0.15
		*V. cholerae*	1.25	0.62
		*V. harveyi*	0.31	0.31
		*V. nigripulchritudo*	1.25	0.15
		*V. parahaemolyticus*	1.25	1.25
		*V. vulnificus*	0.62	0.62
Pseudomonadales	*Acinetobacter*	*A. baumannii*	1.25	2.50
	*Pseudomonas*	*P. aeruginosa*	1.25–2.50	2.50
**FUNGI**				
Saccharomycetales	*Candida*	*C. kefyr*	1.25	1.25
		*C. tropicalis*	0.62–1.25	0.62

-: no MIC found at tested concentrations (from 2.5% to 0.0049%). Screening was performed at least once. MRSA: Methicillin-Resistant *S. aureus*.

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
