# Peer review of "Antimicrobial Spectrum of Titroleaneâ„¢: A New Potent Anti-Infective Agent"

_antibiotics, 2020, doi:10.3390/antibiotics9070391_

Round 1

Reviewer 1 Report

The manuscript is interesting reporting novel data about the antibacterial and antifungal activity of Titroleane a new anti-infective agent obtained from Tea Tree oil.

However, the manuscript is not well written presenting a lot of lacks that need to be clearly addressed before it becomes publishable in a high standard journal. My mainly concerns are related to M&M and result sections and are reported in the following:

Introduction: the rationale of the study should be clearly addressed.

Material and method: usually all the methods employed should be reported in the materials and method section and the results should be related the reported methods. It is not clear to me why the authors reported results relating to "Homogeneity between batches" without describing the materials and methods of how these batches were obtained. In addition, the antimicrobial efficacy results of these batches have been obtained using only three bacteria species. Which was the criterion of this choice? Please add all the info in the M&M section.

The authors include in their experiments, M furfur. Being this species lipid-dependent, a specific CLSI BMD method should be employed to test the antifungal efficacy of tested drugs. Please consider that a reference method has not yet been developed for these yeast species and the culture media, inoculum sizes, incubation times, and the criteria used to determine MIC endpoints differ among studies. I think that the CLSIBMD procedure for M furfur should be clearly reported end /or referenced.

Finally, the procedures for determining MIC must be performed at least three times on different days to obtain evaluable results.

The discussion section has been well written but no data about the low susceptibility of tested strains against chemical drugs has been reported as well as the clinical evolution of the caused diseases. I think that this info should be added in order to justify the usefulness of alternative drugs to control or to prevent the recurrence of their infections.

The minor comments have been detailed in the following.

Results.

Page 2 line  47: “Comparative compounds of standard TTO and Titroleane”:  in the Material and methods there is no capitation with this title. Please make uniform

Page 2 line 69:  Homogeneity between batches: No section in M&M.

Page 3 line 681:”MIC of Titroleane™ and Fraction 2” No section in M&M:

Pages 3 and 4: lines 92- 118: The subsections “2.4. Antibacterial spectrum of Titroleane™” and   “2.5 Activity of Titroleane™ versus TTO” and Tables 4 and 5 should be merged.

Table 5: Please check table 5 accurately.  In table 5 some fungal species disappeared!!!!!!

Material and Methods.

In this section, the methods for the evaluation of the impact of the composition of each extract on plastic containers and the evaluation of antifungal activity of different batches and fraction 2 against only three bacteria spp should be herein added and justify. In addition, info about M. fufur antifungal susceptibility test should be herein reported.  

Page 9, lines 245-249: The different preparation of batches should herein reported.

Page 9, line 276: “Growth condition” should be better changed into “Growth condition and CLSI BMD inoculum preparation.”

Page 10, line 286: the Title of this subsection should be deleted.

Author Response

Reviewer1

The manuscript is interesting reporting novel data about the antibacterial and antifungal activity of Titroleane a new anti-infective agent obtained from Tea Tree oil.

However, the manuscript is not well written presenting a lot of lacks that need to be clearly addressed before it becomes publishable in a high standard journal. My mainly concerns are related to M&M and result sections and are reported in the following:

Introduction: the rationale of the study should be clearly addressed.

Author response:

We would like to thank reviewer 1. Reviewer 1 is correct, so we modified the abstract and added the following sentence:

Tea tree Oil (TTO) is well known for its numerous good properties but might be also irritating or toxic when used topically or ingested, thus limiting the number of possible applications in Humans

In addition, we draw the attention of reviewer 1 to the sentence:

Lines 43-46: “The aim of the study is to evaluate the antimicrobial spectrum of Titroleane™ and to validate that even with the removal of toxic and irritating molecules (also known for their antimicrobial properties), the activity of Titroleane™ is similar to that of TTO”

in fact, we thought that the rational of the study was clearly stated…

Material and method: usually all the methods employed should be reported in the materials and method section and the results should be related the reported methods. It is not clear to me why the authors reported results relating to "Homogeneity between batches" without describing the materials and methods of how these batches were obtained. In addition, the antimicrobial efficacy results of these batches have been obtained using only three bacteria species. Which was the criterion of this choice? Please add all the info in the M&M section.

Author response:

We would like to thank reviewer 1. Reviewer 1 is correct, so we modified the M&M as follow:

First, we modified the paragraph 4.1.2

4.1.2. TTO fractionation

TTO’s fractionation was performed by distillation. TTO was loaded on a fractionating Vigreux-type column (with finger indentations) of 3 meters high and 10 centimeters of diameter, at 72°C at the head of the column. Distillation of TTO was used to separate the mixture into two fractions, based on their volatilities. The fraction containing sesquiterpenes and alcohols was named Titroleane™, while the fraction mostly composed of monoterpenes was named Fraction 2.

And second, we added a new paragraph 4.1.3

4.1.3. Industrial batches

Four industrial batches of Titroleane™ named B1, B2, B3 and B4 were produced at different times. If not specified, Titroleane™ refers to batch B2. Antimicrobial efficacy of these batches was carried out on an initial set of 3 bacterial species: Escherichia coli as a Gram-negative model, Staphylococcus aureus as a Gram-positive model, and Yersinia enterocolitica as an foodborne model. Assays were performed once immediately after production.

The authors include in their experiments, M furfur. Being this species lipid-dependent, a specific CLSI BMD method should be employed to test the antifungal efficacy of tested drugs. Please consider that a reference method has not yet been developed for these yeast species and the culture media, inoculum sizes, incubation times, and the criteria used to determine MIC endpoints differ among studies. I think that the CLSIBMD procedure for M furfur should be clearly reported end /or referenced.

Author response:

We would like to thank reviewer 1, but we believe we have well specified:

  • the culture media used, line 327: “Sabouraud dextrose agar and broth + sterile olive oil was used to grow furfur (UMIP 1634.86).”
  • the growth conditions, line 296: “…moulds and yeasts were grown at 30°C…”
  • as well as MICs method, lines 334-336: “MICs values were measured using the microdilution broth method according to the Clinical and Laboratory Standards Institute guidelines [74], with modification on broths used to fit with organisms’ requirements for growth.”

However, to be completely “clear” and avoid “misinterpretations”, we propose to add the following sentence: lines 336-339: “The standard CLSI protocol was adapted to fungi as follows: fungi were diluted at 105 CFU.mL-1 in Sabouraud broth (Candida albicans, Candida kefyr, Candida tropicalis, Aspergillus niger) or Sabouraud broth + 20% olive oil (Malassezia furfur). Plates were incubated aerobically at 30°C and analyzed after 48 hours.”

Finally, the procedures for determining MIC must be performed at least three times on different days to obtain evaluable results.

Author response:

We would like to thank reviewer 1, but we disagree.

MICs have been performed three independent times or more only for model bacteria (i.e. Escherichia coli, Pseudomonas aeruginosa, Salmonella typhimurium, Klebsiella pneumoniae, Staphylococcus aureus).

In addition, we noticed that several published papers, including in the journal Antibiotics, show MIC experiments performed only once, and we therefore believe that our experiments are in agreement with similar studies published in a high-level journal as Antibiotics This is for example the case of a very recent paper testing Tea Tree Oil using the MIC CLSI protocols, and published very recently, on June 19th, 2020 (Meroni G, Cardin E, Rendina C, Herrera Millar VR, Soares Filipe JF, Martino PA. In Vitro Efficacy of Essential Oils from Melaleuca Alternifolia and Rosmarinus Officinalis, Manuka Honey-based Gel, and Propolis as Antibacterial Agents Against Canine Staphylococcus Pseudintermedius Strains. Antibiotics (Basel). 2020 Jun 19;9(6). pii: E344. doi: 10.3390/antibiotics9060344. PubMed PMID: 32575376.)

The discussion section has been well written but no data about the low susceptibility of tested strains against chemical drugs has been reported as well as the clinical evolution of the caused diseases. I think that this info should be added in order to justify the usefulness of alternative drugs to control or to prevent the recurrence of their infections.

Author response:

We would like to thank reviewer 1, but we are not sure that we fully understand the comment/question of reviewer 1. The aim of present the study was to characterize the antimicrobial spectrum as well as the toxicity of Titroleane™, a new anti-infective agent obtained from TTO but cleared of its toxic monoterpenes part.

This is a preliminary work, carried out on bacterial and fungal strains from collection. But we think that one of the interests of our study is precisely to have tested the efficacy of Titroleane immediately on a very large panel of microorganisms.

Other experiments are necessary, including the use of clinical strains, the use of antimicrobials-resistant strains… before being able to say that the Titroleane is an promising alternative drug... these works are currently underway...

The minor comments have been detailed in the following.

Results.

Page 2 line 47: “Comparative compounds of standard TTO and Titroleane”: in the Material and methods there is no capitation with this title. Please make uniform

Author response:

We agree, we modified the paragraph 4.1.2.

Page 2 line 69: Homogeneity between batches: No section in M&M.

Author response:

We agree, we added the paragraph 4.1.3.

Page 3 line 681: “MIC of Titroleane™ and Fraction 2” No section in M&M:

Author response:

We agree, we modified the paragraph 4.1.2.

Pages 3 and 4: lines 92- 118: The subsections “2.4. Antibacterial spectrum of Titroleane™” and “2.5 Activity of Titroleane™ versus TTO” and Tables 4 and 5 should be merged.

Author response:

We’re sorry, but this is another series of experiences, and this is what we explained, lines 112-113… the differences observed are otherwise commented...

But, in order to avoid misunderstandings, we propose to modify the sentence line 112-113 as follows: “To compare the activity between Titroleane™ and TTO, another series of MICs is performed on two fungi and thirty-one bacteria representing a wide variety of microorganisms.”

Table 5: Please check table 5 accurately. In table 5 some fungal species disappeared!!!!!!

Author response:

No… fungal species do not disappear, please refer to the response above… This is only another series of experiments, on selected strains…

Material and Methods.

In this section, the methods for the evaluation of the impact of the composition of each extract on plastic containers and the evaluation of antifungal activity of different batches and fraction 2 against only three bacteria spp should be herein added and justify. In addition, info about M. fufur antifungal susceptibility test should be herein reported.

Author response:

We agree, we modified the paragraph 4.1.2 and add the following sentences: “Batches were stored in 10 L or 20 L high density polyethylene containers. Samples were collected in 60 mL polystyrene flasks and stored at 22°C and protected from light. In order to evaluate the impact of the composition on plastic containers, 60 mL polystyrene flasks were observed after 2 months. Batch B2 and associated TTO and fraction 2 were used for the visual observation.”

We have also added the paragraph 4.1.3 “Industrial Batches”.

Finally, with respect to M. furfur, we have proposed the amendments outlined above (see paragraph 4.2.4).

Page 9, lines 245-249: The different preparation of batches should herein reported.

Author response:

The corrections have been done.

Page 9, line 276: “Growth condition” should be better changed into “Growth condition and CLSI BMD inoculum preparation.”

Author response:

The modification has been done.

Page 10, line 286: the Title of this subsection should be deleted.

Author response:

The title “culture media” has been deleted.

Reviewer 2 Report

In this paper by Johansen et al, the authors have characterized the antimicrobial activities of Tritoleane, a fraction of Tea tree oil (TTO) without the toxicity associated with TTO. They have used established methods to test the antibiotic activities of Tritoleane against a large number of diverse microorganisms which may potentially be of therapeutic interest.

The authors need to proofread the manuscript carefully for instance:

line 36 fractionation of SETUBIO “leaded” to…

line 72: consequently “fourth” different….

Lines 341 and 346: was tested “tree” times… ,  was washed “tree” times…

There are many such minor errors throughout the manuscript that need to be corrected.

Author Response

Reviewer 2

In this paper by Johansen et al, the authors have characterized the antimicrobial activities of Tritoleane, a fraction of Tea tree oil (TTO) without the toxicity associated with TTO. They have used established methods to test the antibiotic activities of Tritoleane against a large number of diverse microorganisms which may potentially be of therapeutic interest.

The authors need to proofread the manuscript carefully for instance:

line 36 fractionation of SETUBIO “leaded” to…

line 72: consequently “fourth” different….

Lines 341 and 346: was tested “tree” times… , was washed “tree” times…

There are many such minor errors throughout the manuscript that need to be corrected.

Author response:

We would like to thank reviewer 2. Reviewer 2 is correct, so we read our manuscript carefully. We hope to have corrected all minor errors of this kind. All corrections were highlighted in green.

Round 2

Reviewer 1 Report

The authors have revised the manuscript accordingly my suggestions.

However, I have some doubts about the antifungal procedures for M furfur. On my knowledge the employment of Sab broth with 20% of olive oil is never used in CLSI BMD procedures. Thus, the results could be influenced by the composition of the broth. Since M. furfur has only been tested once, I suggest deleting the strain or adding a sentence relating to the difficulty in testing this yeast using CLSIBMD, in the discussion section.

Author Response

Reviewer 2

The authors have revised the manuscript accordingly my suggestions.

However, I have some doubts about the antifungal procedures for M furfur. On my knowledge the employment of Sab broth with 20% of olive oil is never used in CLSI BMD procedures. Thus, the results could be influenced by the composition of the broth. Since M. furfur has only been tested once, I suggest deleting the strain or adding a sentence relating to the difficulty in testing this yeast using CLSIBMD, in the discussion section.

Reviewer 2 is correct, CLSI recommends the use of RPMI 1640, as culture medium, for MIC determination on fungi.

However, many studies have shown that fungal growth can be modified depending on the fungal strains tested and growth medium used.

In addition, in a recent study by Iatta and colleagues (2014) performed on thirty-six M. furfur isolates, the authors concluded that: “The results of the present study provide evidence on the susceptibility of M. furfur isolates from BSIs and suggest the optimal medium for testing the susceptibility of this species. In particular, a good growth of M. furfur incubated for 48 and 72 h was observed in CUB, SDB and RPMI 1640 (P.0.05); thus, suggesting that the three media might be employed for testing the susceptibility of this yeast species to antifungal agents using the CLSI BMD protocol.

This is why we stated from the outset in our manuscript that:

Lines 334-336: “MICs values were measured using the microdilution broth method according to the Clinical and Laboratory Standards Institute guidelines [74], with modification on broths used to fit with organisms’ requirements for growth.”

But to be more precise, we propose to modify the manuscript by adding the following sentence:

Lines 336-339: “Indeed, some authors have shown that there is no significant differences depending on the culture medium used (Sabouraud dextrose broth (SDB), RPMI and Christensen’s urea broth (CUB)) for the determination of MICs using the broth microdilution (BMD) method recommended by the CLSI [75].”

Obviously, the corresponding reference was added:

Iatta R, Figueredo LA, Montagna MT, Otranto D, Cafarchia C. In vitro antifungal susceptibility of Malassezia furfur from bloodstream infections. J Med Microbiol. 2014;63(Pt 11):1467-1473. doi:10.1099/jmm.0.078709-0

To conclude, we would also like to remind reviewer 2 that DMSO was used for samples preparation (§ 4.1.4) thus avoiding formation of an oil phase that might trap hydrophobic molecules.
